# 5,5,5-Trichloropent-3-en-one as a Precursor of 1,3-Bi-centered Electrophile in Reactions with Arenes in Brønsted Superacid CF_3_SO_3_H. Synthesis of 3-Methyl-1-trichloromethylindenes

**DOI:** 10.3390/molecules27196675

**Published:** 2022-10-07

**Authors:** Ivan A. Shershnev, Irina A. Boyarskaya, Aleksander V. Vasilyev

**Affiliations:** 1Department of Organic Chemistry, Institute of Chemistry, Saint Petersburg State University, Universitetskaya nab., 7/9, Saint Petersburg 199034, Russia; 2Department of Chemistry, Saint Petersburg State Forest Technical University, Institutsky per., 5, Saint Petersburg 194021, Russia

**Keywords:** enones, indenes, Friedel-Crafts reaction, carbocations, triflic acid

## Abstract

Reactions of 5,5,5-trichloropent-3-en-2-one Cl_3_CCH=CHC(=O)Me with arenes in Brønsted superacid CF_3_SO_3_H at room temperature for 2 h–5 days afford 3-methyl-1-trichloromethylindenes, a novel class of indene derivatives. The key reactive intermediate, *O*-protonated form of starting compound Cl_3_CCH=CHC(=OH^+^)Me, has been studied experimentally by NMR in CF_3_SO_3_H and theoretically by DFT calculations. The reaction proceeds through initial hydroarylation of the carbon-carbon double bond of starting CCl_3_-enone, followed by cyclization onto the *O*-protonated carbonyl group, leading to target indenes. In general, 5,5,5-trichloropent-3-en-2-one in CF_3_SO_3_H acts as a 1,3-bi-centered electrophile.

## 1. Introduction

Superelectrophilic activation under the action of strong Brønsted and Lewis acids is a useful tool in organic synthesis, giving access to a variety of compounds [1,2,3,4,5,6,7,8]. Protonation (or coordination) of basic centers of organic molecules in Brønsted (or Lewis) acids affords intermediate highly reactive cationic species. In particular, superelectrophilic activation of conjugated enones consequently gives rise to *O*-protonated and *O*,*C*-diprotonated species. The latter takes part in electrophilic aromatic substitution reactions with arenes (Figure 1a) [9,10,11,12,13,14,15,16,17,18]. The formation of *O*,*C*-diprotonated species from various conjugated enone structures, such as butenones [9,18], indenones [12], cinnamic acids, and their esters and amides [13,14,15,16], was proved experimentally by NMR and theoretically by DFT calculations. It has been shown that these dications are key reactive intermediates in various Friedel–Crafts processes [9,10,11,12,13,14,15,16,17,18].

Based on our recent studies on superelectrophilic activation of electron deficient alkenes [19,20,21], we undertook this study on electrophilic activation of *E*-5,5,5-trichloropent-3-en-2-one **1** (CCl_3_-enone). The presence of two electron withdrawing groups, COMe and CCl_3_, at the carbon-carbon double bond increases its electrophilicity, especially under protonation of the carbonyl oxygen-resulting *O*-protonated species **A** (Figure 1b). The second protonation of C=C bond in cation **A** may be hampered due to the strong acceptor characteristics of substituents C(OH^+^)Me and CCl_3_. However, species **A** possesses enough electrophilicity to react with aromatic nucleophiles.

The main goals of this study were to investigate the protonation of *E*-5,5,5-trichloropent-3-en-2-one **1** by NMR and DFT calculations and study its reactions with arenes under the action of strong Brønsted and Lewis acids.

## 2. Results and Discussion

Protonation of CCl_3_-enone **1** in various Brønsted acids (CH_3_COOH, CF_3_COOH, H_2_SO_4_, CF_3_SO_3_H) was initially investigated by means of NMR. According to ^1^H and ^13^C NMR data, CCl_3_-enone **1** gives stable *O*-protonated form **A** in these acids at room temperature (Table 1, see original spectra in Appendix A). Upon increasing the acidity in the row CH_3_COOH→CF_3_COOH→H_2_SO_4_→CF_3_SO_3_H [1], signals of protons H^3^, H^4^ and carbons C^2^, C^4^ are shifted more and more downfield. The corresponding differences in chemical shifts (∆δ = δ_acid_– δ_CDCl3_) for atoms H^3^, H^4^ and C^2^, C^4^ are gradually increased (Table 1). These data reveal that the positive charge is mainly localized on carbons C^2^ and C^4^ in cation **A**, and both these atoms may act as reactive electrophilic centers in consequent interactions with aromatic nucleophiles.

Then, DFT calculations of cations **A**–**C** derived from the protonation of CCl_3_-enone **1** have been carried out. The thermodynamics of their formation, such as Gibbs energies ΔG_298_ of protonation reactions, energies of HOMO/LUMO, electrophilicity indices ω [22,23], charge distribution, and contribution of atomic orbital into LUMO of species **A**–**C** have been estimated (Table 2, see full data in Appendix A).

The formation of *O*-protonated species **A** is very favorable, as the ΔG_298_ value of the protonation is negative (−35 kJ/mol). Secondly, the protonation of the C=C bond, both onto carbons C^3^ and C^4^, which leads to dications **B** and **C**, is, correspondingly, extremely unfavorable, due to the very high positive values of protonation Gibbs energies (Table 2). Thus, the generation of *O*,*C*-diprotonated species **B** and **C** from CCl_3_-enone **1** is very unlikely; that is, in accordance with NMR data (Table 1). Apart from that, it has been found that dication **B** is extremely unstable. It is spontaneously rearranged into species **B1** via a shift of a chlorine atom.

Calculations show that the largest part of positive charge in species **A** is localized on atom C^2^ (0.66 e). Apart from that, this carbon atom contributes significantly to LUMO by 28%. There are similarities between the charge and orbital factors of the electrophilic properties of carbon C^2^. Contrary to that, carbon C^4^ bears no positive charge (−0.06 e), but it contributes significantly into LUMO by 21% (see LUMO visualization of cation **A** in Table 2). Electrophilic properties of atom C^4^ can be mainly explained by orbital factors.

Reactions of CCl_3_-enone **1** with benzene under the action of various Brønsted and Lewis acids have also been conducted (Table 3). The use of strong Lewis acids AlCl_3_ or AlBr_3_ yields complex mixtures of oligomeric materials (entries 1–3). Reaction in H_2_SO_4_ results in the formation of alcohol **3** as a product of hydration of the carbon-carbon double bond; no reaction with benzene occurs (entry 4). Reaction in Brønsted superacid CF_3_SO_3_H (triflic acid, TfOH) at room temperature for 5 days affords indene **2a** in yields 29% (entry 7). Under other conditions (temperature and time) in CF_3_SO_3_H, the formation of **2a** is unsatisfactory (entries 5, 6, 8, 9), as is the reaction in stronger acid FSO_3_H at a low temperature of −78 °C (entry 10). In weaker acids, CH_3_CO_2_H and CH_3_CO_2_H, the reaction does not take place (entries 11–14). These data reveal that the formation of indene **2a** in CF_3_SO_3_H is accompanied by cationic oligomerization processes, which leads to a decrease in the yield of the target compound. The formation of indene **2a** points out that the starting compound **1** in CF_3_SO_3_H behaves as a precursor of the bi-centered electrophile, with reactive cationic centers on carbons C^2^ and C^4^.

Reactions of CCl_3_-enone **1** with other arenes (*o-, m-, p*-xylenes, pseudocumene, and veratrole) in CF_3_SO_3_H, leading to indenes **2b**–**f**, are presented in Figure 2. These reactions with electron donating arenes take much less time (2 h only) at room temperature compared to the reaction with benzene (5 days, Table 1, entry 7). The yields of target indenes **2b**–**f** are moderate (20–47%) due to secondary cationic oligomerization processes.

However, the same reactions with anisole (methoxybenzene) and 1,3-dimethoxybenzene at room temperature for 2 h furnish compounds **4a,b** as products of hydroarylation of the carbon-carbon double bond of starting CCl_3_-enone **1** (Figure 3). Running these reactions at the higher temperature of 60 °C does not lead to the consequent cyclization of compounds **4a,b** into the corresponding indenes **2**.

The data obtained allow proposing plausible reaction mechanisms for transformations of CCl_3_-enone **1** in Brønsted acids (Figure 4). The formation of compounds **4** reveals that the first interaction of arenes with cation **A** occurs at carbon C^4^ of the latter, leading to species **D**. Hydrolysis of these cations affords compounds **4** (Figure 3). In the case of electron donating aryl groups, cations **D** undergo intramolecular cyclization into species **E**. At this stage of the reaction, carbon C^2^ acts as an electrophilic center. Finally, dehydration of **E** gives rise to indenes **2**. Another reaction pathway takes place in H_2_SO_4_. The interaction of cation **A** with hydrosulfate anion HSO_4_^−^ affords species **F**, which is hydrolyzed into alcohol **3**. We additionally examined the reaction of alcohol **3** with benzene in TfOH to obtain indene **2a**. However, only a mixture of oligomeric materials was obtained, with no target indene **2a**. In general, upon the formation of indenes **2**, starting CCl_3_-enone **1** in CF_3_SO_3_H behaves as a precursor of 1,3-bi-centered electrophilic synthon.

It should be especially emphasized that the development of routes for the synthesis of novel indene derivatives such as compounds **2** is a highly important goal for organic chemistry. Indenes are valuable molecules for medicinal uses [24,25,26]. They are widely exploited as ligands in organometallic chemistry [27,28,29,30,31], as structural units in molecular machines [32] and organic photovoltaics [33].

## 3. Experimental Section

### 3.1. General Information

The NMR spectra of solutions of compounds in CDCl_3_ and in acids (CH_3_COOH, CF_3_COOH, H_2_SO_4_, CF_3_SO_3_H) were recorded on a Bruker 400 spectrometer (Billerica, MA, USA) at 25 °C at 400 and 101 MHz for ^1^H and ^13^C NMR spectra, respectively. The residual proton-solvent peaks CDCl_3_ (δ 7.26 ppm) for ^1^H NMR spectra, and the carbon signals of CDCl_3_ (δ 77.0 ppm) for ^13^C NMR spectra were used as references. NMR spectra in acids were referenced to the signal of CH_2_Cl_2_ added as internal standard: δ 5.30 ppm for ^1^H NMR spectra, and δ 53.52 ppm for ^13^C NMR spectra. HRMS-APCI was carried out using the instruments Bruker maXis HRMS-ESI-QTOF (Billerica, MA, USA). Preparative TLC was performed on silica gel 5−40 μm (Merck Co., Kenilworth, NJ, USA) with petroleum ether or petroleum ether-ethyl acetate mixture elution.

### 3.2. DFT Calculations

All computations were carried out at the DFT/HF hybrid level of theory using hybrid exchange functional B3LYP, by using GAUSSIAN 2009 program packages [34]. The geometries optimization was performed using the 6-311+G(2d,2p) basis set (standard 6-311G basis set added with polarization (d,p) and diffuse functions). Optimizations were performed on all degrees of freedom and solvent-phase optimized structures were verified as true minima with no imaginary frequencies. The Hessian matrix was calculated analytically for the optimized structures in order to prove the location of correct minima and to estimate the thermodynamic parameters. For solvent-phase calculations, the Polarizable Continuum Model (PCM, solvent=water) was used.

### 3.3. Preparation and Characterization of Compounds ***1**–**4***

First, *E*-5,5,5-trichloropent-3-en-2-one 1 was obtained in a yield of 83% according to the procedure shown in the literature [35]. Yellow oil. ^1^H NMR (CDCl_3_, 400 MHz) δ, ppm: 7.06 d (*J* 15 Hz, 1H), 6.62 d (*J* 15 Hz, 1H), 2.41 s (3H). ^13^C NMR (CDCl_3_, 101 MHz) δ, ppm: 196.58, 144.35, 127.95, 92.59, 28.85.

The general procedure for the synthesis of indenes **2**, compounds **3** and **4** from *E*-5,5,5-trichloropent-3-en-2-one 1 and arenes in CF_3_SO_3_H. Solution of compound **1** (50 mg, 0.27 mmol) and arene (1.2 equiv., 0.320 mmol) in 2 mL of CF_3_SO_3_H involved stirring at room temperature for 2 h (or other temperature and time, see Table 3 and Figure 3). Then, the reaction mixture was poured into water (25 mL) and extracted with CH_2_Cl_2_ (3 × 20 mL). Combined extract was washed with water (20 mL), saturated aqueous solution of NaHCO_3_ (10 mL), water again (20 mL), and dried over Na_2_SO_4_. The solvent was distilled off under a reduced pressure. The residue was subjected to preparative TLC using petroleum ether or petroleum ether-ethyl acetate mixtures (20:1, vol.) as eluent.

Reactions under the action of other Brønsted (H_2_SO_4_, FSO_3_H) and Lewis (AlCl_3_ and AlBr_3_, 5 equiv. in 5 mL of benzene) acids were carried out in the same way (Table 1).

3-Methyl-1-trichloromethylinden (**2a**), yield of 29% (Table 3). Yellow oil. ^1^H NMR (CDCl_3_, 400 MHz) δ, ppm: 7.96 d (*J* 7.8 Hz, 1H_arom_), 7.44 d (*J* 7.4 Hz, 1H_arom_), 7.35–7.27 m (2H_arom_), 6.32 br. s. (1H, =CH), 4.49 br. s. (1H, CR_3_H), 2.22 t (*J* 1.8 Hz, 3H, CH_3_). ^13^C NMR (CDCl_3_, 101 MHz) δ, ppm: 146.5, 144.1, 141.0, 128.6, 128.4, 125.8, 125.1, 119.4, 100.0 (CCl_3_), 67.5 (CR_3_H), 13.0 (Me). HRMS-APCI: *m*/*z* calc. C_11_H_9_Cl_3_ [M + H]^+^ 246.9848, found 246.9843.

3,4,7-Trimethyl-1-trichloromethylinden (**2b**), yield of 20% (Figure 2). Yellow oil. ^1^H NMR (CDCl_3_, 400 MHz) δ, ppm: 7.04 d (*J* 7.8 Hz, 1H_arom_), 6.97 d (*J* 7.8 Hz, 1H_arom_), 6.25 br. s. (1H, =CH), 4.50 br. s. (1H, CR_3_H), 2.56 s (3H, C_arom_-CH_3_), 2.54 s (3H, C_arom_-CH_3_), 2.35 t (*J* 1.6 Hz, 3H, C_aliph_-CH_3_). ^13^C NMR (101 MHz, CDCl_3_) δ, ppm: 146.3, 144.3, 132.8, 132.0, 131.2, 130.5, 129.1, 128.8, 101.8 (CCl_3_), 65.7 (CR_3_H), 22.6 (C_apom_-CH_3_), 19.5 (C_apom_-CH_3_), 17.8 (CH_3_). HRMS-ESI: *m*/*z* calc. C_13_H_13_Cl_3_ [M + Ag + CH_3_CN]^+^ 421.9399, found 421.9394.

3,5,6-Trimethyl-1-trichloromethylinden (**2c**), yield of 35% (Figure 2). Yellow oil. ^1^H NMR (CDCl_3_, 400 MHz) δ, ppm: 7.72 s (1H_arom_), 7.11s (1H_arom_), 6.22 br. s. (1H, =CH), 4.43 br. s. (1H, CR_3_H), 2.35 s (6H, C_arom_-CH_3_), 2.19 t (*J* 1.7 Hz, 3H, C_aliph_-CH_3_). ^13^C NMR (CDCl_3_, 101 MHz) δ, ppm: 144.47, 143.88, 138.72, 136.85, 134.05, 127.53, 126.50, 120.70, 100.40 (CCl_3_), 67.28 (CR_3_H), 20.21 (C_arom_-CH_3_), 20.09 (C_arom_-CH_3_), 13.03 (CH_3_). HRMS-APCI: *m*/*z* calc. C_13_H_13_Cl_3_ [M + H]^+^ 275.0161, found 275.0156.

3,5,7-Trimethyl-1-trichloromethylinden (**2d**), yield of 23% (Figure 2). Yellow oil. ^1^H NMR (CDCl_3_, 400 MHz) δ, ppm: 6.94 s (1H), 6.92 s (1H_arom_), 6.29 br. s. (1H, =CH), 4.54 br. s. (1H, CR_3_H), 2.58 s (3H, C_arom_-CH_3_), 2.39 s (3H, C_arom_-CH_3_), 2.15 t (*J* 1.6 Hz, 3H). ^13^C NMR (CDCl_3_, 101 MHz) δ, ppm: 147.8, 144.4, 138.7, 136.56, 134.8, 129.9, 129.8, 118.0, 101.6 (CCl_3_), 66.6 (CR_3_H), 22.8 (C_arom_-CH_3_), 21.3 (C_arom_-CH_3_), 13.0 (CH_3_). HRMS-ESI: *m*/*z* calc. C_13_H_13_Cl_3_ [M + Ag + CH_3_CN]^+^ 421.9399, found 421.9394.

3,4,5,7-Tetramethyl-1-trichloromethylinden (**2e**), yield of 47% (Figure 2). Yellow oil. ^1^H NMR (CDCl_3_, 400 MHz) δ, ppm: 6.90 s (1H_arom_), 6.25 br. s. (1H, =CH), 4.44 br. s. (1H, CR_3_H), 2.52 s (3H, C_arom_-CH_3_), 2.44 s (3H, C_arom_-CH_3_), 2.37 s (*J* 1.6 Hz, 3H, C_aliph_-CH_3_), 2.30 s (3H, C_arom_-CH_3_). ^13^C NMR (CDCl_3_, 101 MHz) δ, ppm: 146.4, 144.4, 138.2, 138.1, 132.1, 131.2, 130.8, 127.9, 102.0 (CCl_3_), 65.1 (CR_3_H), 22.4 (CH_3_), 20.2 (CH_3_), 18.8 (CH_3_), 14.8 (CH_3_). HRMS-APCI: *m*/*z* calc. C_14_H_15_Cl_3_ [M + H]^+^ 289.0318, found 289.0312.

5,6-Dimethoxy-3-methyl-1-trichloromethylinden (**2f**), yield of 28% (Figure 2). Yellow oil. ^1^H NMR (CDCl_3_, 400 MHz) δ, ppm: 7.54 s (1H_arom_), 6.85 s (1H_arom_), 6.21 br. s. (1H, =CH), 4.39 br. s. (1H, CR_3_H), 3.97 s (3H, OCH_3_), 3.94 s (3H, OCH_3_), 2.19 t (*J* 1.8 Hz, 3H, CH_3_). ^13^C NMR (CDCl_3_, 101 MHz) δ, ppm: 149.7, 147.4, 143.6, 139.7, 133.4, 127.2, 109.6, 102.9, 100.3 (CCl_3_), 67.2 (CR_3_H), 56.4 (OCH_3_), 56.1 (OCH_3_), 13.2 (CH_3_). HRMS-APCI: *m*/*z* calc. C_13_H_13_Cl_3_O_2_ [M + H]^+^ 307.0059, found 307.0054.

5,5,5-Trichloro-4-hydroxypentane-2-one (**3**) [36], yield of 75% (Table 3). Yellow oil. ^1^H NMR (CDCl_3_, 400 MHz) δ, ppm: 5.03 d (*J* 9.3 Hz, 1H), 3.44 d (*J* 17.4 Hz, 1H), 3.24 dd (*J* 17.4, 9.3 Hz, 1H), 2.29 c (3H). ^13^C NMR (CDCl_3_, 101 MHz) δ, ppm: 201.9, 100.4, 67.2, 48.4, 30.6.

5,5,5-Trichloro-4-(4-methoxyphenyl)pent-2-one (**4a**), yield of 68% (at room temperature for 2 h), 47% (at 60 °C for 0.5 h) (Figure 3). ^1^H NMR (CDCl_3_, 400 MHz) δ, ppm:7.40 d (*J* 8.8 Hz, 2H_arom_), 6.88 d (*J* 8.8 Hz, 2H_arom_), 4.32 dd (*J* 9.2, 3.5 Hz, 1H), 3.80 s (3H, OCH_3_), 3.41 dd (*J* 17.4, 3.5 Hz, 1H), 3.32 dd (*J* 17.4, 9.2 Hz, 1H), 2.11 s (3H, CH_3_). ^13^C NMR (CDCl_3_, 101 MHz) δ, ppm: 204.2 (C=O), 159.7 (C_arom_-OCH_3_), 131.3 (C_arom_), 128.7 (C_arom_), 113.6 (C_arom_), 103.6 (CCl_3_), 59.7, 55.2, 46.5, 30.6 (CH_3_). HRMS-APCI: *m*/*z* calc. C_12_H_13_Cl_3_O_2_ [M + H]^+^ 295.0054, found 295.0054.

5,5,5-Trichloro-4-(2,4-dimethoxyphenyl)pent-2-one (**4b**), yield of 47% (at room temperature for 2 h), 56% (at 60 °C for 0.5 h) (Figure 3). ^1^H NMR (CDCl_3_, 400 MHz) δ, ppm: 7.39 d (*J* 9.3 Hz, 1H_arom_), 6.55–6.44 m (2H_arom_), 5.02 d (*J* 7.9 Hz, 1H), 3.89 s (3H, OCH_3_), 3.81 s (3H, OCH_3_), 3.38 dd (*J* 16.7, 3.6 Hz, 1H), 3.26 dd (*J* 16.7, 10.2 Hz, 1H), 2.09 s (3H, CH_3_). ^13^C NMR (CDCl_3_, 101 MHz) δ, ppm: 204.6 (C=O), 160.8 (C_arom_-OCH_3_), 159.4 (C_arom_-OCH_3_), 128.8 (C_arom_), 117.81 (C_arom_), 104.4 (C_arom_), 103.9 (C_arom_), 98.8 (CCl_3_), 55.1 (OCH_3_), 55.3 (OCH_3_), 50.7, 46.6, 30.2 (CH_3_). HRMS-APCI: *m*/*z* calc. C_13_H_15_Cl_3_O_3_ [M + H]^+^ 325.0165, found 325.0160.

## 4. Conclusions

A novel method for the synthesis of 3-methyl-1-trichloromethylindenes has been developed based on the reaction of 5,5,5-trichloropent-3-en-2-one with arenes in Brønsted superacid CF_3_SO_3_H. In this transformation, the initial 5,5,5-trichloropent-3-en-2-one in CF_3_SO_3_H behaves as a 1,3-bi-centered electrophile.

## Data Availability

Not applicable.

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
