# Peer review of "5,5,5-Trichloropent-3-en-one as a Precursor of 1,3-Bi-centered Electrophile in Reactions with Arenes in Brønsted Superacid CF3SO3H. Synthesis of 3-Methyl-1-trichloromethylindenes"

_molecules, 2022, doi:10.3390/molecules27196675_

Round 1

Reviewer 1 Report

In the manuscript entitled as “5,5,5-Trichloropent-3-en-one as a Precursor of 1,3-Bi-centered Electrophile in Reactions with Arenes in Brønsted Superacid CF3SO3H. Synthesis of 3-Methyl-1-trichloromethylindenes” the authors describe the reactions of 5,5,5-trichloropent-3-en-2-one with arenes in Brønsted superacid CF3SO3H at room temperature for 2 h - 5 days afford 3-methyl-1-trichloromethylindenes, a novel class of indene derivatives. The key reactive intermediate, O-protonated, form of starting compound, has been studied experimentally by NMR in CF3SO3H and theoretically by DFT calculations. Furthermore, a possible reaction mechanism for the reaction to prepare the products has also been proposed.

The journal of Molecules provides an advanced forum for science of chemistry and all interfacing disciplines. In which, this article is a pertinent study, mainly due to the experimental NMR studies, DFT calculations and a novel class of indene prepared. Therefore, I recommend this manuscript for publication in Molecules after consideration of the issues and corrections pointed out below:

1) In the introduction:

·      I think the authors should put a paragraph about indene derivatives in the introduction. Maybe change the paragraph from the ‘Results and discussion’ section to this one (ref 24-33).

·      In the “General Scheme” (Scheme 1, eq. b), products 2, 3 and 4 should be represented.

2) In the Results and discussion:

·         In the Table 1, CH3COOH, CF3COOH, H2SO4 and CF3SO3H were used to the NMR studies. I think, as comparation, in the study on Table 3, should have the results of the reactions of CCl3-enone 1 with benzene under the action of the same acids.

·         Table 3, entry 4, the compound 3 was obtained in 75%. If anisole were used as arene, under the same conditions, is expected to obtain product 3 or 4?

·         Scheme 2, could the yields of the indenes 2b-f be increased if heating of 60 °C was applied?

·         Is the toluene reactive in these conditions (Scheme 2)?

·         Was not the formation of isomers of the products 2d and 2e observed? (Considering the methyl groups attached to the aromatic ring).

·         The sentence “Increasing the reactions temperature to 60 °C does not result in cyclization of compounds 4a,b into the corresponding indenes 2.” it's a little confusing, please rewrite it.

3)  Experimental part:

·         In the “General information” is written that the reference used for the 1H NMR spectra is CDCl3 (δ 7.26 ppm). Please add this peak number in each spectra of SI.

·         In the “Preparation and characterization of compounds” put the number of compounds in bold form.

·         About the indenes, I don’t understand that CH3 has resulted in a triplet signal in the 1H NMR. Please check this multiplicity and put an expansion of this signal in the spectrum (for each compound).  

4) In the Supplementary:

·         The authors should provide a new NMR spectra of compound 2b and 2e, they have some impurities. 

Author Response

Reviewer 1

In the manuscript entitled as “5,5,5-Trichloropent-3-en-one as a Precursor of 1,3-Bi-centered Electrophile in Reactions with Arenes in Brønsted Superacid CF3SO3H. Synthesis of 3-Methyl-1-trichloromethylindenes” the authors describe the reactions of 5,5,5-trichloropent-3-en-2-one with arenes in Brønsted superacid CF3SO3H at room temperature for 2 h - 5 days afford 3-methyl-1-trichloromethylindenes, a novel class of indene derivatives. The key reactive intermediate, O-protonated, form of starting compound, has been studied experimentally by NMR in CF3SO3H and theoretically by DFT calculations. Furthermore, a possible reaction mechanism for the reaction to prepare the products has also been proposed.

The journal of Molecules provides an advanced forum for science of chemistry and all interfacing disciplines. In which, this article is a pertinent study, mainly due to the experimental NMR studies, DFT calculations and a novel class of indene prepared. Therefore, I recommend this manuscript for publication in Molecules after consideration of the issues and corrections pointed out below:

1) In the introduction:

  • I think the authors should put a paragraph about indene derivatives in the introduction. Maybe change the paragraph from the ‘Results and discussion’ section to this one (ref 24-33).

Answer:

We put the paragraph on importance of indenes at the end of the “Result and discussion” section, since in the beginning if the paper in “Introduction” section we did not know what kind of the reaction product will be formed from compound 1. It is some kind of intrigue of the story in the beginning.

  • In the “General Scheme” (Scheme 1, eq. b), products 23and 4 should be represented.

Answer:

Please, see the answer above. Once again, we wish to keep an intrigue of the story in the beginning of the paper to interest readers.

2) In the Results and discussion:

  • In the Table 1, CH3COOH, CF3COOH, H2SO4and CF3SO3H were used to the NMR studies. I think, as comparation, in the study on Table 3, should have the results of the reactions of CCl3-enone 1 with benzene under the action of the same acids.

Answer:

These data were added to the Table 3, see entries 11-14 and the corresponding explanation in text of the paper.

  • Table 3, entry 4, the compound 3was obtained in 75%. If anisole were used as arene, under the same conditions, is expected to obtain product or 4?

Answer:

This reaction of 1 with anisole in sulfuric acid led to compound 3. It points out that nucleophilicity of hydrosulfane anion HSO3- is higher than aromatic nucleophiles.

  • Scheme 2, could the yields of the indenes 2b-fbe increased if heating of 60 °C was applied?

Answer:

At higher temperature at 60°C, the yields of target indenes 2b-f were less, perhaps, due to secondary cationic oligomerization processes.

  • Is the toluene reactive in these conditions (Scheme 2)?

Answer:

Reaction with toluene gave a complex mixture of regiosomeric indenes due to electrophilic aromatic substitution in ortho- and para-positions of toluene.

  • Was not the formation of isomers of the products 2dand 2e observed? (Considering the methyl groups attached to the aromatic ring).

Answer:

Compound 2d and 2e were isolated as solely isomers.

  • The sentence “Increasing the reactions temperature to 60 °C does not result in cyclization of compounds4a,into the corresponding indenes 2.” it's a little confusing, please rewrite it.

Answer:

This sentence was rephrased. Now, it is:

“Running these reactions at higher temperature at 60°C does not lead to a consequent cyclization of compounds 4a,b into the corresponding indenes 2.”

3)  Experimental part:

  • In the “General information” is written that the reference used for the 1H NMR spectra is CDCl3(δ 7.26 ppm). Please add this peak number in each spectra of SI.

Answer:

Usually, residual proton signals from deuterated solvents are not marked in proton spectra to avoid mixing of these signals with signals from compound.

  • In the “Preparation and characterization of compounds” put the number of compounds in bold form.

Answer:

It was added.

  • About the indenes, I don’t understand that CH3has resulted in a triplet signal in the 1H NMR. Please check this multiplicity and put an expansion of this signal in the spectrum (for each compound).  

Answer:

This triplet comes from spin-spin coupling of this CH3 group with protons H1 (>CH-CCl3) and H2 (=CH-) of the indene system with a small constant 1.6-1.8 Hz. The same weak spin-spin interaction results in broadening of signals of indene protons H1 and H2, that should be quartets of doublets (or quintets).

4) In the Supplementary:

  • The authors should provide new NMR spectra of compound 2b and 2e, they have some impurities. 

Answer:

As we have mentioned in the paper, the formation of indenes 2 is accompanied by the formation of products of cationic oligomerization. These oligomers have very close chromatographic retention parameters to indenes 2. In some cases, it was very difficult to separate indenes 2 from non-polar admixtures of oligomers, even using chromatographic eluation by non-polar hexanes.

Reviewer 2 Report

This is an interesting paper containing an account of the development of an intramolecular cyclisation reaction. of trichloromethyl-substituted enones using strong acid catalysis.

In terms of the synthetic chemistry, using TfOH, cyclisation products (2) are formed in moderate yields of 20-47%, but in two cases the same conditions lead to a non-cyclised addition product in slightly better yields. Under other conditions, a beto-hydroxy ketone was formed, i.e. with a weaker acid. These results are what might be expected with this range of reagents and conditions.

The authors condition an investigation by NMR which reveals increasing NMR shifts as stronger acid is added. Although I am happy that an initial O-protonated species is formed, I am no convinced that a second protonation leads to the O,C-diprotonated species.  A DFT study did indicate some support for this although the energy seems rather high and the positive charges are in different positions in Scheme 1 compared to Table 2.

There is also discussion of a Cl shift reaction, which seems a reasonable possibility, but this does not seem to appear in the experimental results.

The addition and cycloaddition reactions could take place, from a mechanistic point of view, from the monoprotonated species A (or the O-protonated species above, which is just a different canonical form of the same cation.

It is possible be that the NMR shifts reflect the formation of an increasing proportion of a monoprotonated species in the mixture rather than a dicationic species.  

the work is coupled with some molecular modelling which appears to be of good quality however I am not an expert on this area so an expert opinion is required on the computational work.

The experimental work is of reasonable quality with appropriate characterisation provided for new compounds and NMR spectra in the SI (acceptable, although not excellent).

Overall, I feel that the experimental results are interesting and, whilst not a major breakthrough, will be an interesting contribution to the journal. In regard to the synthetic work, I am happy to support publication. I’d like the authors to clarify whether they are confident that dication is actually formed compared to a larger proportion of a monocation, and also whether the monocation or dication (if they know) are specifically responsible for the cyclisation reactions. However an expert opinion is require on the computational part of the work.

Author Response

Reviewer 2

This is an interesting paper containing an account of the development of an intramolecular cyclisation reaction. of trichloromethyl-substituted enones using strong acid catalysis.

In terms of the synthetic chemistry, using TfOH, cyclisation products (2) are formed in moderate yields of 20-47%, but in two cases the same conditions lead to a non-cyclised addition product in slightly better yields. Under other conditions, a beto-hydroxy ketone was formed, i.e. with a weaker acid. These results are what might be expected with this range of reagents and conditions.

The authors condition an investigation by NMR which reveals increasing NMR shifts as stronger acid is added. Although I am happy that an initial O-protonated species is formed, I am no convinced that a second protonation leads to the O,C-diprotonated species.  A DFT study did indicate some support for this although the energy seems rather high and the positive charges are in different positions in Scheme 1 compared to Table 2.

There is also discussion of a Cl shift reaction, which seems a reasonable possibility, but this does not seem to appear in the experimental results.

The addition and cycloaddition reactions could take place, from a mechanistic point of view, from the monoprotonated species A (or the O-protonated species above, which is just a different canonical form of the same cation.

It is possible be that the NMR shifts reflect the formation of an increasing proportion of a monoprotonated species in the mixture rather than a dicationic species.  

the work is coupled with some molecular modelling which appears to be of good quality however I am not an expert on this area so an expert opinion is required on the computational work.

The experimental work is of reasonable quality with appropriate characterisation provided for new compounds and NMR spectra in the SI (acceptable, although not excellent).

Overall, I feel that the experimental results are interesting and, whilst not a major breakthrough, will be an interesting contribution to the journal. In regard to the synthetic work, I am happy to support publication. I’d like the authors to clarify whether they are confident that dication is actually formed compared to a larger proportion of a monocation, and also whether the monocation or dication (if they know) are specifically responsible for the cyclisation reactions. However an expert opinion is require on the computational part of the work.

Answer:

Experimental data point out that, the most probably, the key reactive intermediate is O-protonated form A generated from enone 1.

Round 2

Reviewer 1 Report

In the manuscript entitled as “5,5,5-Trichloropent-3-en-one as a Precursor of 1,3-Bi-centered Electrophile in Reactions with Arenes in Brønsted Superacid CF3SO3H. Synthesis of 3-Methyl-1-trichloromethylindenes” the authors describe the reactions of 5,5,5-trichloropent-3-en-2-one with arenes in Brønsted superacid CF3SO3H at room temperature for 2 h - 5 days afford 3-methyl-1-trichloromethylindenes, a novel class of indene derivatives.

I believe that the present study is in accordance with the profile of the journal, that the manuscript is well written and after all changes and explanations performed by the authors, I have seen an improvement in the quality of the manuscript. Thus, I consider this article accepted for publication in Molecules.